# Near Instance-Optimal PAC Reinforcement Learning for Deterministic MDPs

**Andrea Tirinzoni**[*]
Meta AI
Paris, France
tirinzoni@fb.com

**Aymen Al-Marjani**
UMPA, ENS Lyon
Lyon, France
aymen.al_marjani@ens-lyon.fr

**Emilie Kaufmann**
Univ. Lille, CNRS, Inria, Centrale Lille, UMR 9189 - CRIStAL
Lille, France
emilie.kaufmann@univ-lille.fr

## Abstract

In probably approximately correct (PAC) reinforcement learning (RL), an agent is required to identify an $\varepsilon$-optimal policy with probability $1 - \delta$. While minimax optimal algorithms exist for this problem, its instance-dependent complexity remains elusive in episodic Markov decision processes (MDPs). In this paper, we propose the first nearly matching (up to a horizon squared factor and logarithmic terms) upper and lower bounds on the sample complexity of PAC RL in deterministic episodic MDPs with finite state and action spaces. In particular, our bounds feature a new notion of sub-optimality gap for state-action pairs that we call the deterministic return gap. While our instance-dependent lower bound is written as a linear program, our algorithms are very simple and do not require solving such an optimization problem during learning. Their design and analyses employ novel ideas, including graph-theoretical concepts (minimum flows) and a new maximum-coverage exploration strategy.

## 1 Introduction

In reinforcement learning [RL, 40], an agent interacts with an environment modeled as a Markov decision process (MDP) by sequentially selecting actions and receiving feedback in the form of reward signals. Depending on the application, the agent may seek to maximize the cumulative rewards received during learning (which is typically phrased as a *regret minimization* problem) or to minimize the number of learning interactions (i.e., the *sample complexity*) for identifying a near-optimal policy. The latter *pure exploration* problem was introduced in [21] under the name of Probably Approximately Correct (PAC) RL: given two parameters $\varepsilon, \delta > 0$, the agent must return a policy that is $\varepsilon$-optimal with probability at least $1 - \delta$. Our work focuses on this problem in the context of episodic (a.k.a. finite-horizon) tabular MDPs.

The PAC RL problem has been mostly studied under the lens of minimax (or worst-case) optimality. In the episodic setting, the algorithm proposed in [12] has sample complexity bounded by $O(SAH^2 \log(1/\delta)/\varepsilon^2)$ for an MDP with $S$ states, $A$ actions, horizon $H$, and time-homogeneous transitions and rewards (i.e., not depending on the stage). This is minimax optimal for such a context [11]. Similarly, in [35] the authors designed a strategy with $O(SAH^3 \log(1/\delta)/\varepsilon^2)$ complexity in time-inhomogeneous MDPs, which was later shown to be minimax optimal [17].

---

[*]Work done while at Inria Lille.

While the minimax framework provides a strong notion of statistical optimality, it does not account for one of the most desirable properties for an RL algorithm: the ability to adapt to the difficulty of the MDP instance. For this reason, researchers recently started to investigate the instance-dependent complexity of PAC RL. Earlier attempts were made in the simplified setting where the agent has access to a generative model (i.e., it can query observations from any state-action pair using a simulator) in $\gamma$-discounted infinite-horizon MDPs [50, 5]. The online setting, where the agent can only sample trajectories from the environment, has been studied in [4] for discounted MDPs and in [46] for episodic time-inhomogeneous MDPs. All these works derive sample complexity bounds that scale with certain gaps between optimal value functions. For instance, in the episodic setting, the *value gap* $\Delta_h(s,a) := V_h^\star(s) - Q_h^\star(s,a)$[2] intuitively characterizes the degree of sub-optimality of action $a$ for state $s$ at stage $h$. Unfortunately, these bounds are known to be sub-optimal and how to achieve instance optimality remains one of the main open questions. In fact, recent works on regret minimization [42, 13] showed that value gaps are often overly conservative, and the same holds for PAC RL. We refer the reader to Appendix A for a deeper discussion on problem-dependent results in RL and the review of other related PAC learning frameworks.

The main challenge towards instance optimality is that existing lower bounds for exploration problems in MDPs [5, 42, 4, 13] are written in terms of non-convex optimization problems. Their "implicit" form makes it hard to understand the actual complexity of the setting and, thus, to design optimal algorithms. Existing solutions either derive explicit *sufficient* complexity measures that inspire algorithmic design [46], or solve (a relaxation of) the optimization problem from the lower bound using the empirical MDP as a proxy for the unknown MDP [4]. The latter extends the Track-and-Stop idea originally proposed in [22] for bandits ($H = 1$), and requires in particular a large amount of forced exploration. Both solutions have limitations. On the one hand, it is not clear if and how such sufficient complexity measures or relaxations are related to an actual lower bound. On the other hand, strategies solving a black-box optimization problem to find an optimal exploration strategy are typically very inefficient and often come with either only asymptotic ($\delta \to 0$) guarantees or with poor (far from minimax optimal) sample complexity in the regime of moderate $\delta$.

**Contributions** This paper presents a complete study of PAC RL in tabular *deterministic* episodic MDPs with time-inhomogeneous transitions, a sub-class of stochastic MDPs where state transitions are deterministic and the agent observes stochastic rewards from unknown distributions. Our first contribution is an *instance-dependent lower bound* on the sample complexity of any PAC algorithm. We show that the number of visits $n_h^\tau(s,a)$ to any state-action-stage triplet $(s,a,h)$ at the stopping time $\tau$ satisfies

$$\mathbb{E}[n_h^\tau(s,a)] \gtrsim \frac{\log(1/\delta)}{\max(\overline{\Delta}_h(s,a), \varepsilon)^2}, \tag{1}$$

where $\overline{\Delta}_h(s,a) := V_1^\star - \max_{\pi \in \Pi_{s,a,h}} V_1^\pi$, with $V_1^\pi$ the expected return of policy $\pi$, $V_1^\star$ the optimal expected return, and $\Pi_{s,a,h}$ the set of all deterministic policies that visit $(s,a)$ at stage $h$. We call these quantities the *deterministic return gaps* due to their closeness with the *return gaps* introduced in [13] for general MDPs. In deterministic MDPs, the deterministic return gaps are actually $H$ times larger than the return gaps and they are never smaller than value gaps. Our lower bound on the sample complexity $\tau$ is then the value of a *minimum flow* with local lower bounds (1), i.e., roughly the minimum number of policies that must be played to ensure (1) for all $(s,a,h)$. To our knowledge, this is the first instance-dependent lower bound for the PAC setting in episodic MDPs.

On the algorithmic side, we design EPRL, a *generic elimination-based method* for PAC RL, and couple it with a novel adaptive sampling rule called *maximum-coverage sampling*. The latter is a simple strategy which does not require solving the optimization problem from the lower bound at learning time in a Track-and-Stop fashion. Instead, it greedily selects the policy that maximizes the number of visited under-sampled triplets $(s,a,h)$, i.e. those having received the least amount of visits so far. We prove that EPRL is $(\varepsilon, \delta)$-correct under any sampling rule. Moreover, we show that the sample complexity of EPRL with max-coverage sampling matches our instance-dependent lower bound up to logarithmic factors and a multiplicative $O(H^2)$ term, while also being minimax optimal. Finally, we perform numerical simulations on random deterministic MDPs which reveal that EPRL can indeed improve over existing minimax-optimal algorithms tailored for the deterministic case.

---

[2] $V^\star$ and $Q^\star$ respectively denote the optimal value and action-value functions, that are defined in Section 2.

## 2 Preliminaries

Let $\mathcal{M} := (\mathcal{S}, \mathcal{A}, \{f_h, \nu_h\}_{h \in [H]}, s_1, H)$ be a *deterministic* time-inhomogeneous finite-horizon MDP, where $\mathcal{S}$ is a finite set of $S$ states, $\mathcal{A}$ is a finite set of $A$ actions, $f_h : \mathcal{S} \times \mathcal{A} \to \mathcal{S}$ and $\nu_h : \mathcal{S} \times \mathcal{A} \to \mathcal{P}(\mathbb{R})$ are respectively the transition function and the reward distribution at stage $h \in [H]$, $s_1 \in \mathcal{S}$ is the unique initial state, and $H$ is the horizon. Without loss of generality, we assume that, at each stage $h \in [H]$ and state $s \in \mathcal{S}$, only a subset $\mathcal{A}_h(s) \subseteq \mathcal{A}$ of actions is available. We denote by $r_h(s, a) := \mathbb{E}_{x \sim \nu_h(s,a)}[x]$ the expected reward after taking action $a$ in state $s$ at stage $h$.

A deterministic policy $\pi = \{\pi_h\}_{h \in [H]}$ is a sequence of mappings $\pi_h : \mathcal{S} \to \mathcal{A}$. We let $\Pi := \{\pi \mid \forall h \in [H], s \in \mathcal{S} : \pi_h(s) \in \mathcal{A}_h(s)\}$ be the set of all valid deterministic policies. Executing a policy $\pi \in \Pi$ on MDP $\mathcal{M}$ yields a deterministic sequence of states and actions $(s_h^\pi, a_h^\pi)_{h \in [H]}$, where $s_1^\pi = s_1$, $a_h^\pi = \pi_h(s_h^\pi)$ for all $h \in [H]$, and $s_h^\pi = f_{h-1}(s_{h-1}^\pi, a_{h-1}^\pi)$ for all $h \in \{2, \dots, H\}$. We let $\mathcal{S}_h := \{s \in \mathcal{S} \mid \exists \pi \in \Pi : s_h^\pi = s\}$ be the subset of states that are reachable at stage $h \in [H]$. Finally, we define $N := \sum_{h=1}^H \sum_{s \in \mathcal{S}_h} |\mathcal{A}_h(s)|$ as the total number of reachable state-action-stage triplets.

For each $(s, a, h)$, the *action-value function* $Q_h^\pi(s, a)$ of a policy $\pi \in \Pi$ quantifies the expected return when starting from $s$ at stage $h$, playing $a$ and following $\pi$ thereafter. In deterministic MDPs, it has the simple expression $Q_h^\pi(s, a) = r_h(s, a) + V_{h+1}^\pi(f_h(s, a))$, where $V_h^\pi(s) := Q_h^\pi(s, \pi_h(s))$ is the corresponding value function (with $V_{H+1}^\pi(s) = 0$). The expected *return* of $\pi$ is simply its value at the initial state, i.e., $V_1^\pi(s_1) = \sum_{h=1}^H r_h(s_h^\pi, a_h^\pi)$. We let $\Pi^\star := \{\pi^\star \in \Pi : V_1^{\pi^\star}(s_1) = \max_{\pi \in \Pi} V_1^\pi(s_1)\}$ be the set of *optimal policies*, i.e., those with maximal return. Finally, we denote by $V_h^\star(s)$ and $Q_h^\star(s, a)$ the optimal value and action-value function, respectively. These are related by the Bellman optimality equations as $Q_h^\star(s, a) = r_h(s, a) + V_{h+1}^\star(f_h(s, a))$ and $V_h^\star(s) = \max_{a \in \mathcal{A}_h(s)} Q_h^\star(s, a)$.

**Learning problem** The agent interacts with an MDP $\mathcal{M}$ in episodes indexed by $t \in \mathbb{N}$. At the beginning of the $t$-th episode, the agent selects a policy $\pi^t \in \Pi$ based on past history through its *sampling rule*, executes it on $\mathcal{M}$, and observes the corresponding deterministic trajectory $(s_h^{\pi^t}, a_h^{\pi^t})_{h \in [H]}$ together with random rewards $(y_h^t)_{h \in [H]}$, where $y_h^t \sim \nu_h(s_h^{\pi^t}, a_h^{\pi^t})$. At the end of each episode, the agent may decide to terminate the process through its *stopping rule* and return a policy $\widehat{\pi}$ prescribed by its *recommendation rule*. We denote by $\tau$ its random stopping time. An algorithm for PAC identification is thus made of a triplet $(\{\pi^t\}_{t \in \mathbb{N}}, \tau, \widehat{\pi})$. The goal of the agent is two-fold. First, for given parameters $\varepsilon, \delta > 0$, it must return an $\varepsilon$-optimal policy with probability at least $1 - \delta$.

**Definition 1.** *An algorithm is $(\varepsilon, \delta)$-PAC on a set of MDPs $\mathfrak{M}$ if, for all $\mathcal{M} \in \mathfrak{M}$, it stops a.s. with*

$$\mathbb{P}_{\mathcal{M}}\left(V_1^{\widehat{\pi}}(s_1) \geq V_1^\star(s_1) - \varepsilon\right) \geq 1 - \delta.$$

Second, it should stop as early as possible, i.e., by minimizing the *sample complexity* $\tau$. Henceforth, we assume that the transition function $f$ is known but not the reward distribution $\nu$. Note that if the transitions are unknown, the agent can still estimate them (since it knows that $\mathcal{M}$ is deterministic) with at most $N \leq SAH$ episodes.

**Minimum flows** We review some basic concepts from graph theory which will be at the core of our algorithms and analyses later. Full details can be found in Appendix B. First note that a deterministic MDP (without reward) can be represented as a *directed acyclic graph* (DAG) with one arc for each available state-action-stage triplet. Let $\mathcal{E} := \{(s, a, h) : h \in [H], s \in \mathcal{S}_h, a \in \mathcal{A}_h(s)\}$ be the set of arcs in the DAG. The minimum flow problem, originally introduced in [45] and later studied in, e.g., [1, 2, 10], consists of finding a flow (i.e., an allocation of visits) of minimal value which satisfies certain demand constraints in each arc of the graph. In our specific setting, we define a *flow* as any non-negative function $\eta : \mathcal{E} \to [0, \infty)$ that belongs to the following set

$$\Omega := \left\{\eta : \mathcal{E} \to [0, \infty) \mid \sum_{(s', a') : f_{h-1}(s', a') = s} \eta_{h-1}(s', a') = \sum_{a \in \mathcal{A}_h(s)} \eta_h(s, a) \ \forall h > 1, s \in \mathcal{S}_h\right\}.$$

This implies that a flow, seen as an allocation of visits to the arcs, satisfies the *navigation constraints* (i.e., incoming and outcoming flows are equal at each state). The minimum flow for a non-negative *lower-bound* function $\underline{c} : \mathcal{E} \to [0, \infty)$ is the solution to the following linear program (LP):

$$\varphi^\star(\underline{c}) := \min_{\eta \in \Omega} \sum_{a \in \mathcal{A}_1(s_1)} \eta_1(s_1, a) \quad \text{s.t.} \quad \eta_h(s, a) \geq \underline{c}_h(s, a) \quad \forall (s, a, h) \in \mathcal{E}.$$

Intuitively, the goal is to minimize the amount of flow leaving the initial state while satisfying the navigation and demand constraints. We note that more efficient algorithms exist for this problem than the LP formulation, e.g., the variant of the Ford-Fulkerson method proposed in [10] which is guaranteed to find an integer solution when the lower bound function is integer-valued.

## 3    The Complexity of PAC RL in Deterministic MDPs

Before stating our lower bound, we formally introduce the new notion of sub-optimality gap it features and compare it with other notions that appeared in the literature.

**On sub-optimality gaps**    The most popular notion of sub-optimality gap is the so-called *value gap*. It was introduced first in the discounted infinite-horizon setting [e.g., 50] and later for episodic MDPs [e.g., 38, 48]. Formally, in the latter context, the value gap of any action $a \in \mathcal{A}_h(s)$ in state $s \in \mathcal{S}_h$ at stage $h \in [H]$ is $\Delta_h(s, a) := V_h^\star(s) - Q_h^\star(s, a)$. Such a notion of gap appears in the complexity measure for PAC RL proposed in [46]. In the deterministic setting, such a complexity measure can be written as $\mathcal{C}(\mathcal{M}, \varepsilon) = \sum_{(s,a,h)} \frac{1}{\max(\widetilde{\Delta}_h(s,a),\varepsilon)^2}$, where $\widetilde{\Delta}_h(s, a) = \min_{a':\Delta_h(s,a')>0} \Delta_h(s, a')$ if $a$ is the unique optimal action at $(s, h)$, and $\widetilde{\Delta}_h(s, a) = \Delta_h(s, a)$ otherwise. Intuitively, the (inverse) value gap is proportional to the difficulty of learning whether an action $a$ is sub-optimal for state $s$ at stage $h$. Then, $\mathcal{C}(\mathcal{M}, \varepsilon)$ is proportional to the difficulty of learning a near optimal action at *all* states and stages. Recent works [42, 13] showed that this is actually not necessary: if one only cares about computing a policy maximizing the return at the initial state, it is not necessary to learn an optimal action at states which are not visited by such an optimal policy, in particular when the return of all policies visiting the state is small. The *return gap* [13] was introduced to cope with this limitation. In deterministic MDPs, it can be expressed as $\overline{\mathrm{gap}}_h(s, a) := \frac{1}{H} \min_{\pi \in \Pi_{s,a,h}} \sum_{\ell=1}^{h} \Delta_\ell(s_\ell^\pi, a_\ell^\pi)$, where we denote by $\Pi_{s,a,h} := \{\pi \in \Pi : s_h^\pi = s, a_h^\pi = a\}$ the subset of deterministic policies that visit $(s, a)$ at stage $h$. In words, the return gap of $(s, a, h)$ is proportional to the *sum* of value gaps along the best trajectory (i.e., one with maximal return) that visits $(s, a)$ at stage $h$. Intuitively, this means that, if $\Delta_h(s, a)$ is extremely small but all policies visiting $(s, a)$ at stage $h$ need to play a highly sub-optimal action before, then $\Delta_h(s, a) \ll \overline{\mathrm{gap}}_h(s, a)$. In the deterministic case, our lower bound reveals that the normalization by $H$ is not necessary, and we define the *deterministic return gap* to be

$$\overline{\Delta}_h(s, a) := V^\star(s_1) - \max_{\pi \in \Pi_{s,a,h}} V^\pi(s_1). \tag{2}$$

Using the well-known relationship $V_1^\star(s_1) - V_1^\pi(s_1) = \sum_{h=1}^{H} \Delta_h(s_h^\pi, a_h^\pi)$ [e.g., 42, Proposition 5], it is easy to see that $\Delta_h(s, a) \leq \overline{\Delta}_h(s, a) = H \times \overline{\mathrm{gap}}_h(s, a)$.

**Lower Bound**    We now present our instance-dependent lower bound based on deterministic return gaps, which will guide us in the design and analysis of sample efficient algorithms. This result is the first instance-dependent lower bound for PAC RL in the episodic setting. Lower bounds for $\varepsilon$-best arm identification in a bandit model (which corresponds to $H = S = 1$) were derived in [34, 14, 23], while problem-dependent regret lower bounds for finite-horizon MDPs are provided in [13, 42].

We consider the class $\mathfrak{M}_{\sigma^2}$ of deterministic MDPs with $\sigma^2$-*Gaussian* rewards, in which $\nu_h(s, a) = \mathcal{N}(r_h(s, a), \sigma^2)$. Let $\Pi^\varepsilon := \{\pi \in \Pi : V_1^\pi(s_1) \geq V_1^\star(s_1) - \varepsilon\}$ be the set of all $\varepsilon$-optimal policies and denote by $\mathcal{Z}_h^\varepsilon := \{s \in \mathcal{S}_h, a \in \mathcal{A}_h(s) : \Pi_{s,a,h} \cap \Pi^\varepsilon \neq \emptyset\}$ the set of state-action pairs that are reachable at stage $h$ by some $\varepsilon$-optimal policy. Note that $\overline{\Delta}_h(s, a) \leq \varepsilon$ for all $(s, a) \in \mathcal{Z}_h^\varepsilon$.

**Theorem 1.** *Let $\sigma^2 > 0$ and fix any MDP $\mathcal{M} \in \mathfrak{M}_{\sigma^2}$. Then, any algorithm which is $(\varepsilon, \delta)$-PAC on the class $\mathfrak{M}_{\sigma^2}$ must satisfy, for any $h \in [H]$, $s \in \mathcal{S}_h$, and $a \in \mathcal{A}_h(s)$,*

$$\mathbb{E}_{\mathcal{M}}[n_h^\tau(s, a)] \geq \underline{c}_h(s, a) := \frac{\sigma^2 \log(1/4\delta)}{4 \max(\overline{\Delta}_h(s, a), \overline{\Delta}_{\min}^h, \varepsilon)^2}, \tag{3}$$

*where $\overline{\Delta}_{\min}^h := \min_{(s',a'):\overline{\Delta}_h(s',a')>0} \overline{\Delta}_h(s', a')$ if $|\mathcal{Z}_h^\varepsilon| = 1$ and $\overline{\Delta}_{\min}^h := 0$ otherwise. Moreover, for $\underline{c} : \mathcal{E} \to [0, \infty)$ the lower bound function defined above,*

$$\mathbb{E}_{\mathcal{M}}[\tau] \geq \varphi^\star(\underline{c}). \tag{4}$$

The first lower bound (3) is on the number of visits required for any state-action-stage triplet. It intuitively shows that an $(\varepsilon, \delta)$-PAC algorithm must visit each triplet proportionally to its inverse

deterministic return gap. The second one (4) shows that the actual sample complexity of the algorithm must be at least the value of a minimum flow computed with the local lower bounds (3), i.e. that the algorithm must play the minimum number of episodes (i.e., policies) that guarantees (3) for each $(s, a, h)$. Intuitively, due to the navigation constraints of the MDP, there might be no algorithm which tightly matches (3) for each $(s, a, h)$, and (4) is exactly enforcing these constraints. While $\varphi^\star(\underline{c})$ has no explicit form, Lemma 6 in Appendix B gives an idea of how it scales with the gaps:

$$\max_{h \in [H]} \sum_{s \in \mathcal{S}_h} \sum_{a \in \mathcal{A}_h(s)} \frac{\sigma^2 \log(1/4\delta)}{4 \max(\overline{\Delta}_h(s,a), \overline{\Delta}_{\min}^h, \varepsilon)^2} \leq \varphi^\star(\underline{c}) \leq \sum_{h \in [H]} \sum_{s \in \mathcal{S}_h} \sum_{a \in \mathcal{A}_h(s)} \frac{\sigma^2 \log(1/4\delta)}{4 \max(\overline{\Delta}_h(s,a), \overline{\Delta}_{\min}^h, \varepsilon)^2}.$$

Observe that the quantity on the right-hand side resembles the complexity measure $\mathcal{C}(\mathcal{M}, \varepsilon)$ [46], except that value gaps are replaced by return gaps. This implies that, in general, our lower bound can be much smaller than this complexity. For instance, in an MDP with extremely small value gaps in states which are not visited by an optimal policy, $\varphi^\star(\underline{c})$ does not scale with such gaps at all.

In Appendix C.2 we further provide a minimax lower bound for PAC RL in deterministic MDPs scaling as $\Omega\left(SAH^2 \log(1/\delta)/\varepsilon^2\right)$, with a reduced $H^2$ dependency compared to the $H^3$ that appear in the stochastic case [17]. We note that faster rates for deterministic MDPs have already been obtained in other RL settings [e.g., 49]. The BPI-UCRL algorithm [29] particularized to deterministic MDPs is matching this lower bound and is thus minixal optimal. We now present the first algorithm which is simultaneously minimax optimal for deterministic MDPs and nearly matching (up to $O(H^2)$ and logarithmic factors) the lower bound of Theorem 1.

## 4 EPRL and Max-Coverage Sampling

We propose a general Elimination-based scheme for PAC RL, called EPRL (Algorithm 1). At each episode $t \in \mathbb{N}$, the algorithm plays a policy $\pi^t$ selected by some sampling rule. Then, based on the collected samples, the algorithm updates its statistics and eliminates all actions which are detected as sub-optimal with enough confidence. This procedure is repeated until a stopping rule triggers.

Formally, EPRL maintains an estimate $\hat{r}_h^t(s,a) := \frac{1}{n_h^t(s,a)} \sum_{l=1}^t y_h^l \mathbb{1}\left(s_h^l = s, a_h^l = a\right)$, with $\hat{r}_h^0 = 0$, of the unknown mean reward $r_h(s,a)$ for each $(s,a,h)$. Here $n_h^t(s,a) := \sum_{l=1}^t \mathbb{1}\left(s_h^l = s, a_h^l = a\right)$ is the number of times $(s,a)$ is visited at stage $h$ up to episode $t$. We define the following upper and lower confidence intervals to the value functions of a policy $\pi \in \Pi$:

$$\overline{Q}_h^{t,\pi}(s,a) := \hat{r}_h^t(s,a) + b_h^t(s,a) + \overline{V}_{h+1}^{t,\pi}(f_h(s,a)), \quad \overline{V}_h^{t,\pi}(s) := \overline{Q}_h^{t,\pi}(s, \pi_h(s)),$$

$$\underline{Q}_h^{t,\pi}(s,a) := \hat{r}_h^t(s,a) - b_h^t(s,a) + \underline{V}_{h+1}^{t,\pi}(f_h(s,a)), \quad \underline{V}_h^{t,\pi}(s) := \underline{Q}_h^{t,\pi}(s, \pi_h(s)),$$

where $b_h^t(s,a)$ is a *bonus function*, i.e., the width of the confidence interval at $(s,a,h)$. We assume that rewards are $\sigma^2$-sub-Gaussian with a known factor $\sigma^2$,[3] which allows us to choose

$$b_h^t(s,a) := \sqrt{\frac{\beta(n_h^t(s,a), \delta)}{n_h^t(s,a)}}, \quad \beta(t, \delta) := 2\sigma^2 \log\left(\frac{4t^2 N}{\delta}\right). \tag{5}$$

**Elimination rule**  Algorithm 1 keeps a set of active (or candidate) actions $\mathcal{A}_h^t(s)$ for each stage $h \in [H]$, state $s \in \mathcal{S}_h$, and episode $t \in \mathbb{N}$. Let $\Pi^t := \{\pi \in \Pi \mid \forall s, h : \pi_h(s) \in \mathcal{A}_h^t(s) \vee \mathcal{A}_h^t(s) = \emptyset\}$ be the subset of *active* policies that only play active actions at episode $t$. Note that an active policy can play an arbitrary action in states where all actions have been eliminated. As can be seen in Line 7 of Algorithm 1, action $a$ is eliminated from $\mathcal{A}_h^t(s)$ if $\max_{\pi \in \Pi_{s,a,h} \cap \Pi^{t-1}} \overline{V}_1^{t,\pi}(s_1) \leq \max_{\pi \in \Pi} \underline{V}_1^{t,\pi}(s_1)$, that is, when we are confident that none of the policies visiting $(s,a)$ at stage $h$ is optimal. We recall that $\Pi_{s,a,h}$ denotes the set of all deterministic policies that visit $s, a$ at stage $h$. The maximum restricted to $\Pi_{s,a,h}$ can be easily computed by standard dynamic programming (e.g., it is enough to set the reward to $-\infty$ for all state-action pairs different than $(s,a)$ at stage $h$). If $\Pi_{s,a,h} \cap \Pi^{t-1} = \emptyset$, we set the maximum to $-\infty$ so that the elimination rule triggers.

**Remark 1.** *While defining $\Pi^t$ simplifies the presentation, EPRL neither stores nor enumerates the set of active policies. In particular, EPRL does not eliminate policies but rather $(s,a,h)$ triplets. The sets $\mathcal{A}_h^t(s)$ can be updated in polynomial time by dynamic programming without ever computing $\Pi^t$.*

---

[3]Note that sub-Gaussianity generalizes the common assumption of bounded rewards in $[0, 1]$ (in which case $\sigma^2 = 1/4$) and the one of Gaussian rewards with variance $\sigma^2$ (as used in the lower bound of Theorem 1).

---
**Algorithm 1** Elimination-based PAC RL (EPRL) for deterministic MDPs
---
1: **Input:** deterministic MDP (without reward) $\mathcal{M} := (\mathcal{S}, \mathcal{A}, \{f_h\}_{h \in [H]}, s_1, H), \varepsilon, \delta$
2: Initialize $\mathcal{A}_h^0(s) \leftarrow \mathcal{A}_h(s)$ for all $h \in [H], s \in \mathcal{S}_h$
3: Set $n_h^0(s,a) \leftarrow 0$ for all $h \in [H], s \in \mathcal{S}_h, a \in \mathcal{A}_h(s)$
4: **for** $t = 1, \ldots$ **do**
5:     Play $\pi^t \leftarrow \text{SAMPLINGRULE}()$
6:     Update statistics $n_h^t(s,a), \hat{r}_h^t(s,a)$
7:     $\mathcal{A}_h^t(s) \leftarrow \mathcal{A}_h^{t-1}(s) \cap \left\{ a \in \mathcal{A} : \max_{\pi \in \Pi_{s,a,h} \cap \Pi^{t-1}} \overline{V}_1^{t,\pi}(s_1) \geq \max_{\pi \in \Pi} \underline{V}_1^{t,\pi}(s_1) \right\}$
8:       where $\Pi^{t-1} \leftarrow \left\{ \pi \in \Pi \mid \forall s, h : \pi_h(s) \in \mathcal{A}_h^{t-1}(s) \vee \mathcal{A}_h^{t-1}(s) = \emptyset \right\}$ (need not be stored/computed)
9:     **if** $\max_{\pi \in \Pi^t} \left( \overline{V}_1^{\pi,t}(s_1) - \underline{V}_1^{\pi,t}(s_1) \right) \leq \varepsilon$ or $\forall h \in [H], s \in \mathcal{S}_h : |\mathcal{A}_h^t(s)| \leq 1$ **then**
10:       Stop and recommend $\hat{\pi} \in \arg\max_{\pi \in \Pi^t} \overline{V}_1^{\pi,t}(s_1)$
11:     **end if**
12: **end for**

13: **function** MAXCOVERAGE()
14:     Let $k_t \leftarrow \min_{h \in [H], s \in \mathcal{S}_h, a \in \mathcal{A}_h^{t-1}(s)} n_h^{t-1}(s,a) + 1$ and $\bar{t}_{k_t} \leftarrow \inf_{l \in \mathbb{N}} \{l : k_l = k_t\}$
15:     **if** $t \bmod 2 = 1$ **then**
16:       **return** $\pi^t \leftarrow \arg\max_{\pi \in \Pi} \sum_{h=1}^{H} \mathbb{1}\left( a_h^\pi \in \mathcal{A}_h^{\bar{t}_{k_t}-1}(s_h^\pi), n_h^{t-1}(s_h^\pi, a_h^\pi) < k_t \right)$
17:     **else**
18:       **return** $\pi^t \leftarrow \arg\max_{\pi \in \Pi^{t-1}} \sum_{h=1}^{H} b_h^{t-1}(s_h^\pi, a_h^\pi)$          (MAXDIAMETER)
19:     **end if**
---

**Stopping rule** EPRL uses two different stopping rules (Line 9). The first one checks whether, for all active policies $\pi \in \Pi^t$, the confidence interval on the return, $\overline{V}_1^{\pi,t}(s_1) - \underline{V}_1^{\pi,t}(s_1) = 2\sum_{h=1}^{H} b_h^t(s_h^\pi, a_h^\pi)$, which we refer to as *diameter*, is below $\varepsilon$. The second one checks whether each set $\mathcal{A}_h^t(s)$ contains either 1 action or 0 actions (which happens when the state is unreachable by an optimal policy). In both cases, we recommend the optimistic (active) policy (Line 10).

**Sampling rule** While EPRL may be used with different sampling rules, we recommend the max-coverage sampling rule described in Algorithm 1. This sampling rule aims at ensuring that no $(s, a, h)$ triplet remains under-visited for too long. This is achieved by selecting the policy which greedily maximizes the number of visited under-sampled triplets, denoted by $\mathcal{U}_t$. The quantity $k_t = \min_{(s,a,h):a \in \mathcal{A}_h^{t-1}(s)} n_h^{t-1}(s,a) + 1$ can be interpreted as the target minimum number of visits from active triplets that we want to achieve in round $t$ and permit to define

$$\pi^t = \arg\max_{\pi \in \Pi} \sum_{h=1}^{H} \mathbb{1}\left((s_h^\pi, a_h^\pi, h) \in \mathcal{U}_t\right) \text{ with } \mathcal{U}_t = \left\{ (s,a,h) : a \in \mathcal{A}_h^{\bar{t}_{k_t}-1}(s), n_h^{t-1}(s,a) < k_t \right\},$$

where $\bar{t}_k = \inf\{t : k_t = k\}$ is the first round in which the target is set to $k$. The argmax over $\Pi$ can be computed using dynamic programming. We emphasize that this argmax is not restricted to the set of active policies, meaning that we may play eliminated actions in order to augment the coverage (that is, the minimal number of visits) faster. Every even round, max-coverage instead chooses an active policy maximizing the diameter featured in the stopping rule (max-diameter sampling). As we shall see in our analysis, this dichotomous behavior is needeed in order to maintain minimax-optimality.

**Comparison with other elimination-based algorithms** The work of [19] provides a heuristic using action eliminations to find an $\varepsilon$-optimal policy in a discounted MDP. However, no sample complexity guarantees are given for this algorithm, which uses a different elimination rule, based on confidence intervals on the optimal value function, and a uniform sampling rule. The MOCA algorithm [46] also uses a different action elimination rule compared to ours. In particular, the decision to eliminate $(s, a, h)$ is made based only on rewards that can be obtained after visiting $(s, a, h)$. Moreover, this algorithm uses a complex phase-based sampling rule, while the sampling rule of EPRL is fully adaptive.

# 5 Theoretical Guarantees

Our first result, proved in Appendix D.2.1, shows that EPRL is $(\varepsilon, \delta)$-PAC under any sampling rule. It follows from the fact that 1) the choice of bonus function (5) ensures that all the confidence intervals are valid and 2) state-action pairs from optimal trajectories are never eliminated when this holds.

**Theorem 2.** *Algorithm 1 is $(\varepsilon, \delta)$-PAC provided that the sampling rule makes it stop almost surely.*

We now analyze the sample complexity of EPRL combined with max-coverage sampling.

**Theorem 3.** *(Informal version of Theorem 8 in Appendix D.3) With probability at least $1 - \delta$, the sample complexity of EPRL combined with the maximum-coverage sampling rule satisfies $\tau = \widetilde{O}(\varphi^\star(g))$, where $g : \mathcal{E} \to [0, \infty)$ is the lower bound function defined by*

$$g_h(s, a) := \frac{32\sigma^2 H^2}{\max\left(\overline{\Delta}_h(s, a), \overline{\Delta}_{\min}, \varepsilon\right)^2} \left( \log\left(\frac{4N^3}{\delta}\right) + 8 \log\left( \frac{16\sigma H \log\left(\frac{4N^3}{\delta}\right)}{\max\left(\overline{\Delta}_h(s, a), \overline{\Delta}_{\min}, \varepsilon\right)} \right) \right) + 2.$$

*Moreover, with the same probability, $\tau = \widetilde{O}(\frac{SAH^2}{\varepsilon^2} \log(1/\delta))$, where $\widetilde{O}$ hides logarithmic terms.*

First note that EPRL combined with such a sampling rule is *minimax optimal*, since it matches the worst-case lower bound derived in Appendix C.2. In addition, the leading term in the instance-dependent complexity is the value of a minimum flow with a lower bound function $g$ that, in case multiple disjoint optimal trajectories exist[4], matches the gap-dependence in (1). If we suppose that there exist at least two disjoint optimal trajectories, in which case $\overline{\Delta}_{\min} = \overline{\Delta}_{\min}^h = 0$, then, thanks to Lemma 7 in Appendix B, one can easily see that $\varphi^\star(g) \leq \alpha H^2 \varphi^\star(\underline{c}) + \varphi^\star(g')$, where $g'_h(s, a) := \widetilde{O}(H^2 / \max\left(\overline{\Delta}_h(s, a), \varepsilon\right)^2)$ does not depend on $\delta$, $\underline{c}$ is the "optimal" lower bound function from (1), and $\alpha$ is a numerical constant. Hence, in the asymptotic regime ($\delta \to 0$), $\varphi^\star(g)$ matches our lower bound up to a $O(H^2)$ multiplicative factor.

**Remark 2.** *Since Theorem 1 was derived for Gaussian rewards, EPRL is instance-optimal only when the reward distribution is Gaussian. This is not surprising since it is well known from the bandit literature [e.g., 32] that sample complexity bounds scaling with a sum of inverse squared gaps are optimal only for Gaussian distributions. Note, however, that EPRL works in greater generality and achieves complexity $\varphi^\star(\underline{c})$ for any $\sigma^2$-sub-Gaussian distribution without knowing its specific form (e.g., whether it is Gaussian or not). What is the optimal rate for other common distributions (e.g., bounded rewards in $[0, 1]$) and how to achieve it remains an open question.*

Finally, our sample complexity bound has an extra multiplicative logarithmic term which roughly scales as $O(\log(H) \log(H \log(1/\delta)/\varepsilon))$. While this term makes the dependence on $\delta$ sub-optimal by a $\log \log(1/\delta)$ factor, we show in Appendix E that it can be removed in the specific case of tree-based MDPs [13].

**Remark 3.** *We believe that the sub-optimality on $H$ could be reduced to a single $H$ factor by boosting the lower bound. In Appendix E, we show that this is indeed possible in tree-based MDPs. As for the upper bound, reducing $H^2$ to $H$ is likely to require tighter concentration bounds on values.*

**Remark 4.** *In Appendix D.4, we prove that, when using the max-diameter sampling rule (Line 18 in Algorithm 1) at each step, the sample complexity is $\widetilde{O}(\sum_{(s,a,h)} H^2 / \max(\overline{\Delta}_h(s, a), \overline{\Delta}_{\min}, \varepsilon)^2)$. While this scales with the same gaps as Theorem 3, it is only a naive upper bound to the minimum flow value (see Section 3). The intuition is that max-diameter sampling alone does not ensure that all triplets are visited sufficiently often, which prevents us from tightly controlling their elimination times.*

**Proof sketch** The complete proof is given in Appendix D.3. It first relies on the following crucial result which relates the deterministic return gaps to the sum of confidence bonuses.

**Lemma 1** (Diameter vs gaps). *With probability at least $1 - \delta$, for any $t \in \mathbb{N}, h \in [H], s \in \mathcal{S}_h, a \in \mathcal{A}_h(s)$, if $a \in \mathcal{A}_h^t(s)$ and the algorithm did not stop at the end of episode $t$,*

$$\max\left(\frac{\overline{\Delta}_h(s, a)}{4}, \frac{\overline{\Delta}_{\min}}{4}, \frac{\varepsilon}{2}\right) \leq \max_{\pi \in \Pi^{t-1}} \sum_{h=1}^{H} b_h^t(s_h^\pi, a_h^\pi),$$

---

[4]When there is a unique optimal trajectory, our upper bound scales with $\overline{\Delta}_{\min} = \min_{h \in [H]} \overline{\Delta}_{\min}^h$ at all stages $h$, while the lower bound scales with $\overline{\Delta}_{\min}^h$ at stage $h$. We believe the latter should be improvable to obtain a dependence on $\overline{\Delta}_{\min}$ matching the one in the upper bound.

where $\overline{\Delta}_{\min} := \min_{h\in[H]} \min_{s\in\mathcal{S}_h} \min_{a:\overline{\Delta}_h(s,a)>0} \overline{\Delta}_h(s,a)$ *if there exists a unique optimal trajectory* $(s_h^\star, a_h^\star)_{h\in[H]}$, *and* $\overline{\Delta}_{\min} := 0$ *in the opposite case.*

In our analysis, we refer to the set of consecutive time steps $\{t \in \mathbb{N} : k_t = k\}$ as the $k$-th *period*. Using the fact that in period $k+1$ each active triplet has been visited at least $k$ times (which allows to upper bound each bonus $b_h^t(s_h^\pi, a_h^\pi)$ for $\pi \in \Pi^{t-1}$ by a quantity scaling in $\sqrt{1/k}$), one can use Lemma 1 to obtain an upper bound $\overline{\kappa}_{s,a,h} \simeq \frac{H^2 \log(1/\delta)}{\max(\overline{\Delta}_h(s,a),\overline{\Delta}_{\min},\varepsilon)^2}$ on the last period in which $(s,a,h)$ is active (Lemma 18 in Appendix D.3). A crucial step of the proof is then to upper bound the duration of the $k$-th period, $d_k := \sum_{t=1}^{\tau} \mathbb{1}(k_t = k)$.

**Lemma 2.** $d_k \leq 2(\log(H)+1)\varphi^\star(\underline{c}^k)$ *where* $\underline{c}_h^k(s,a) = \mathbb{1}(a \in \mathcal{A}_h^{\bar{t}_k-1}(s), n_h^{\bar{t}_k-1}(s,a) < k)$.

The intuition behind this result is as follows. Recall that the goal of the max-coverage sampling rule in period $k$ is to visit at least once each $(s,a,h)$ that is active (i.e., $a \in \mathcal{A}_h^{\bar{t}_k-1}(s)$) and undersampled (i.e., $n_h^{\bar{t}_k-1}(s,a) < k$). By definition, the minimum flow $\varphi^\star(\underline{c}^k)$ is the minimum number of policies that need to be played to achieve this goal. Interestingly, Lemma 2 shows that the number of policies played by max-coverage to visit all active undersampled triplets is very close to its theoretical minimum, despite the fact that the algorithm never computes an actual minimum flow. We prove this by interpreting max-coverage sampling as a greedy maximization of some coverage function (related to a minimum flow problem) and leveraging the theory of sub-modular maximization [e.g., 31].

Thanks to Lemma 2, we have that

$$\tau \leq 2(\log(H)+1)\sum_{k=1}^{k_\tau} \varphi^\star(\underline{c}^k),$$

where $k_\tau$ is the index of the period at which the algorithm stops. To bound this quantity we carefully apply the theory of minimum flows and their dual problem of *maximum cuts*. Let us define a cut $\mathcal{C}$ as any subset of states containing the initial state and let $\mathcal{E}(\mathcal{C})$ be the set of arcs that connect states in $\mathcal{C}$ with states not in $\mathcal{C}$. The well-known min-flow-max-cut theorem (Theorem 4 stated in Appendix B) states that, for any lower bound function $\underline{c}$, $\varphi^\star(\underline{c}) = \max_{\mathcal{C}\in\mathfrak{C}} \sum_{(s,a,h)} \underline{c}_h(s,a)$, where $\mathfrak{C}$ denotes the set of all valid cuts. Then,

$$k\varphi^\star(\underline{c}^k) \leq \max_{\mathcal{C}\in\mathfrak{C}} \sum_{(s,a,h)\in\mathcal{E}(\mathcal{C})} k\mathbb{1}\left(a \in \mathcal{A}_h^{\bar{t}_k-1}(s)\right) \leq \max_{\mathcal{C}\in\mathfrak{C}} \sum_{(s,a,h)\in\mathcal{E}(\mathcal{C})} (\overline{\kappa}_{s,a,h}+1) = \varphi^\star(g),$$

where $g : \mathcal{E} \to [0,\infty)$ is defined by $g_h(s,a) = \overline{\kappa}_{s,a,h}+1$. It follows that

$$
\begin{aligned}
\tau \quad &\leq \quad 2(\log(H)+1)\sum_{k=1}^{k_\tau} \frac{1}{k}\varphi^\star(g)\\
&\leq \quad 2(\log(H)+1)\left(\log(k_\tau)+1\right)\varphi^\star(g)\\
&\leq \quad 2(\log(H)+1)\left(\max_{(s,a,h)}\log(\overline{\kappa}_{s,a,h})+1\right)\varphi^\star(g).
\end{aligned}
$$

Using the expression of $\overline{\kappa}_{s,a,h}$ given in Lemma 18 of Appendix D.3 concludes the proof of the stated $\widetilde{O}(\varphi^\star(g))$ instance-dependent bound. For the worse-case bound, we refer the reader to Theorem 12.

$\square$

# 6 Experiments

We compare numerically EPRL to the minimax optimal BPI-UCRL algorithm [29], adapted to the deterministic setting, on synthetic MDP instances. For EPRL, we experiment with two sampling rules: max-coverage (maxCov) and max-diameter (maxD, see Line 18 of Algorithm 1). We defer to Appendix F some implementation details, including a precise description of the BPI-UCRL baseline.

We generate random "easy" deteministic MDP instances with Gaussian rewards of variance 1 using the following protocol. For fixed $S, A, H$ the mean rewards $r_h(s,a)$ are drawn i.i.d. from a uniform

distribution over $[0, 1]$ and for each state-action pair, the next state is chosen uniformly at random in $\{1, \ldots, S\}$. Finally, we only keep MDP instances whose minimum value gap, denoted by $\Delta_{\min}$, is larger than 0.1. Our first observation is that depending on the MDP, the identity of the best performing algorithm can be different. In Figure 1 we show the distribution of the sample complexity (estimated over 10 Monte Carlo simulations) for three different MDPs obtained from our sampling procedure with $S, A = 2$ and $H = 3$ and for algorithms that are run with parameters $\delta = 0.1$ and $\varepsilon = 1.5\Delta_{\min}$.

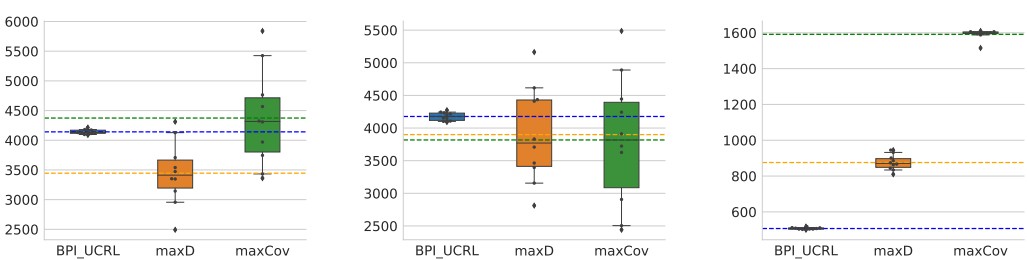

Figure 1: Distribution of stopping times on particular MDPs over 10 runs, with $\varepsilon = 1.5\Delta_{\min}$. The horizontal lines represent the average sample complexity.

To get a better understanding of this phenomenon, we then generated 10 MDP instances of size $(S, A, H) = (2, 2, 3)$ and for each MDP we ran EPRL and BPI-UCRL for 25 values of $\varepsilon$ in a grid $[0.05\Delta_{\min}, 10\Delta_{\min}]$ and $\delta = 0.1$. We ran 10 Monte-Carlo simulations for each value of the triplet (MDP, algorithm A, $\varepsilon$), in order to estimate the expected sample complexity $\mathbb{E}_A[\tau_\delta]$. In Figure 2 we plot the relative performance (ratio of sample complexities) of different algorithms as a function of the value of $\varepsilon/\Delta_{\min}$: each point corresponds to a different MDP and a different value of $\varepsilon$. We observe that for large values of $\varepsilon/\Delta_{\min}$, BPI-UCRL has a smaller sample complexity than both versions of EPRL, with a ratio never exceeding 2 (resp. 3) for max-diameter (resp. max-coverage). However, in the more interesting small $\varepsilon/\Delta_{\min}$ regime EPRL is better by several orders of magnitude. This is expected since, for small $\varepsilon$, EPRL is able, through its elimination rule, to identify the optimal policy long before the diameter goes below $\varepsilon$. We observe that the threshold of $\varepsilon/\Delta_{\min}$ at which EPRL algorithms become a better choice than BPI-UCRL seems to vary with the MDP.

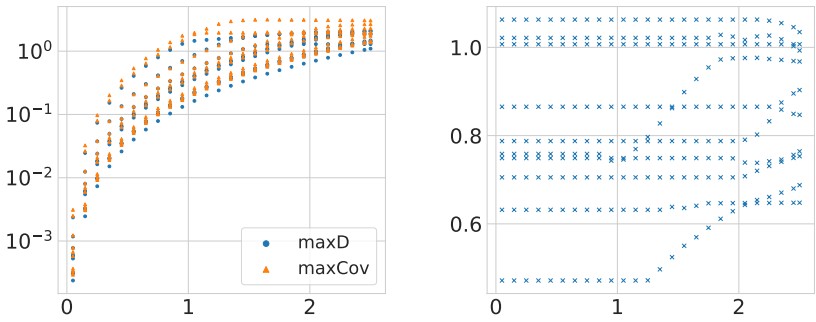

Figure 2: Ratios in log-scale $\mathbb{E}_A[\tau_\delta]/\mathbb{E}_{\text{BPI-UCRL}}[\tau_\delta]$ for A in $\{\text{maxD}, \text{maxCov}\}$ (left) and $\mathbb{E}_{\text{maxD}}[\tau_\delta]/\mathbb{E}_{\text{maxCov}}[\tau_\delta]$ (right) as a function of $\varepsilon/\Delta_{\min}$.

Our experiments also reveal an intriguing phenomenon: the use of max-diameter sampling within EPRL often outperforms max-coverage sampling, even if there exists MDPs (2 out of 10 in our experiments) in which max-coverage is indeed empirically better. We leave as future work to obtain a better characterizations of MDPs for which EPRL with max-coverage sampling performs best.

# 7 Discussion

We derived an instance-dependent and a worst-case lower bound characterizing the complexity of PAC RL in deterministic MDPs, and proposed a general elimination algorithm together with a novel maximum-coverage sampling rule that nearly matches them (up to $O(H^2)$ and logarithmic factors). We conclude with some discussion about our results and future directions.

**Max-coverage vs max-diameter**    While minimax optimality can be easily achieved with very simple strategies (like max-diameter or BPI-UCRL), instance optimality requires careful algorithmic design. Our coverage-based strategy is built around the idea of "uniformly" exploring the whole MDP, while using an elimination strategy to ensure that no $(s, a, h)$ is sampled much more than what the lower bound prescribes. Notably, this sampling rule is very simple, while exiting PAC RL algorithms with instance-dependent complexity are all quite involved [46, 4]. Moreover, max-coverage sampling naturally extends to stochastic MDPs, e.g., by doing optimistic planning on an MDP with a reward function equal to 1 for under-sampled triplets and 0 for the others. Finally, in our experiments on random instances, we observed that max-diameter is often comparable or better than max-coverage. We leave as future work to investigate whether the latter is also provably near instance-optimal.

**Computational aspects**    Our sampling rule requires solving one dynamic program per episode, which takes $O(N)$ time. The bottleneck is the elimination rule, which requires $O(N^2)$ per-episode time complexity to solve one dynamic program for each active triplet. However, we note that eliminations could be checked periodically (e.g., even at exponentially-separated times) without significantly compromising the sample complexity guarantees.

**Improving our results**    Our instance-dependent upper bound for max-coverage sampling is sub-optimal by a factor $H^2$ and a multiplicative $O(\log\log(1/\delta))$ term. In Appendix E, we show that, for the specific sub-class of tree-based MDPs [13], we can obtain improved results in all these aspects. In particular, we show that (1) the lower bound scales with an extra factor $H$ and it is fully explicit, (2) the multiplicative log terms in the sample complexity of coverage-based sampling can be removed, and (3) maximum-diameter sampling also achieves near instance-optimal guarantees.

**Beyond Gaussian distributions**    As it is common, e.g., in the bandit literature, the gaps in our lower and upper bounds are optimal only for Gaussian reward distributions. Extending Theorem 1 to general distributions is actually simple (see, e.g., [28] and Lemma 8 in Appendix C). However, this would yield gaps written in terms of KL divergences between arm distributions rather than differences of mean rewards as in the Gaussian case. How to match such gaps is an interesting open question.

**Instance optimality in stochastic MDPs**    The main open question is how to achieve (near) instance-optimality for PAC RL in stochastic MDPs. We believe that many of the results presented in this paper could help in this direction. First, our instance-dependent lower bound could be extended to the stochastic case by modifying return gaps to include visitation probabilities and minimum flows to account for stochastic navigation constraints. Second, on the algorithmic side, our maximum-coverage sampling rule easily extends to stochastic MDPs as mentioned above, while our elimination rule could also be adapted by computing the optimistic return of policies visiting a certain $(s, a, h)$ with a least some probability, which corresponds to a constrained MDP problem [e.g., 18]. Studying how these components behave in stochastic MDPs is an exciting direction for future work.

## Acknowledgments and Disclosure of Funding

Aymen Al-Marjani ackowledges the support of the Chaire SeqALO (ANR-20-CHIA-0020). Emilie Kaufmann acknoweldges the support of the French National Research Agency under the BOLD project (ANR-19-CE23-0026-04).

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
