# OpenReview forum: "Near Instance-Optimal PAC Reinforcement Learning for Deterministic MDPs"
_NeurIPS.cc/2022/Conference — NeurIPS 2022 Accept_

### Official Review · Reviewer_vy1w · 2022-07-11

**Rating:** 5
**Confidence:** 4
**Soundness:** 3 good
**Presentation:** 3 good
**Contribution:** 2 fair

**Summary:**

This work studies the PAC RL problem in the tabular finite horizon setting for deterministic transition MDPs. The paper derives instance dependent lower bounds and an algorithm which nearly matches the lower bounds.

**Questions:**

Given the discussion regarding computing a minimum policy cover through min-flow/max-cut (Algorithm 2) and the fact that once all paths to a state-action pair are eliminated, e.g., by eliminating all incomming edges into the state-action pair, will Algorithm 3 ever select a policy which visits the state-action pair with no incomming edges? Further, will Algorithm 1 play a policy which visits such a state-action pair?

Do the authors think they can derive a meaningful lower bound, maybe in terms of a min-flow LP, for the setting of bounded rewards?



**Limitations:**

This work is theoretical in nature and future societal impacts are hard to judge.

**Strengths And Weaknesses:**

Strengths:

1.) Originality -- the paper shows some interesting results in terms of relating the MaxCoverage function in Algorithm 1, to the StaticMaxCoverage in Algorithm 3. In particular the observation that the MaxCoverage function behaves similarly to computing a minimum cover of the remaining (not eliminated) state-actions pairs and playing according to this cover (as done in the StaticMaxCoverage) is interesting and could potentially lead to imporved PAC RL algorithms for more general settings.

2.) Quality -- the paper is technically sound and provides some interesting new ideas as mentioned above. Further, the paper improves on prior work lower bounds by a polynomial factor of the horizon.

3.) Clarity and significance -- the paper is well written and mostly clear and easy to understand.

Weaknesses:

1.) Originality -- while the authors claim to propose a new notion of sub-optimality gap, this notion already exists in prior work up to a normalization factor of horizon. The novel contributions of the paper seem to mostly be in terms of removing spurious factors of horizon in upper and lower bounds from prior work. It is worth mentioning that the lower bound stated in Theorem 1 can not be directly compared to regret minimization lower bounds in prior work as the regret minimization lower bounds are asymptotic and regret minimization and PAC RL are not really aligned in their objectives as pointed out by Wagenmaker et al. 2022.

2.) Clarity and significance -- I do not find the claims of matching upper and lower bounds true. It is important to note that Theorem 1 holds in a different setting from Theorem 3, that is in Theorem 1 there are no assumptions made about boundedness of the rewards, and in fact this is a very important part of the proof, making the result similar in spirit to the lower bound in Theorem 4.5  (Dann et al. 2021). The results in Theorem 3, in my understanding, hold for the more standard setting of bounded rewards. It is important to note that there could be a large gap between the two settings, making the claims of matching upper and lower bounds inaccurate. In particular, for the bounded rewards setting, there exist MDPs in which state-action pairs need not be visited even once for to be eliminated. This is due to the fact, that it is possible to eliminate all actions on paths to such state-action pairs before ever visiting such state-action pairs and hence there is no alternative MDP as in Lemma 8 of the current paper.

Given that the paper does not really establish lower bounds in the more interesting setting of bounded rewards and the upper bounds mostly seem to remove horizon dependent factors, I am unsure of how strong the contributions of this work are.

References:

Wagenmaker, Andrew J., Max Simchowitz, and Kevin Jamieson. "Beyond no regret: Instance-dependent pac reinforcement learning." Conference on Learning Theory. PMLR, 2022.

Dann, Christoph, et al. "Beyond value-function gaps: Improved instance-dependent regret bounds for episodic reinforcement learning." Advances in Neural Information Processing Systems 34 (2021): 1-12.

---

> ### Author Response · Authors · 2022-08-01
> **Answer to Reviewer vy1w**
>
> We thank the reviewer for the detailed feedback. We answer all the questions below.
>
> > while the authors claim to propose a new notion of sub-optimality gap, this notion already exists in prior work up to a normalization factor of horizon
>
> The reviewer is right. Indeed our deterministic return gaps are the same as the return gaps of Dann et al. 2021 up to a multiplicative factor of $H$, as we clarify in the paper. We stress that the main novelty here is in the fact that we show, through nearly matching upper and lower bounds, that this is the right notion of gap for PAC RL (with Gaussian rewards).
>
> > The novel contributions of the paper seem to mostly be in terms of removing spurious factors of horizon in upper and lower bounds from prior work.
>
> We respectfully disagree on this point. Our paper provides several novel contributions which go beyond improving factors of the horizon and which seem to have been overlooked by the reviewer:
>
> 1. First, no existing instance-dependent lower bound was known for PAC RL. All the available lower bounds for regret minimization (eg, those of Dann et al. 2021 and Tirinzoni et al. 2021) are written in terms of intractable/non-convex optimization problems. The most related to ours is Theorem 4.5 of Dann et al. 2021 which considers deterministic MDPs with Gaussian rewards, though our proof for PAC RL departs from theirs in different aspects (we elaborate on this point below).
>
> 2. On the algorithmic side, the only prior work on instance-dependent PAC RL that  we are aware of is the one of Wagenmaker et al. 2022. They propose a rather involved algorithm for general stochastic MDPs whose sample complexity scales with value gaps. For the specific setting of deterministic MDPs, we improve over their work along multiple dimensions, including the gap dependence, factors of $H$, low order terms, and simplicity of the proposed strategy.
>
> 3. Beyond instance-dependent results, we also derive a minimax lower bound for deterministic MDPs and show that our algorithm matches it, confirming that deterministic MDPs are easier than stochastic ones even in the worst case.
>
> 4. Finally, we achieve all this with novel analytical tools connecting exploration to minimum flows and maximum cuts, which we believe to be of independent interest and could lead to improved analyses of existing/new algorithms as the reviewer suggests.
>
> > I do not find the claims of matching upper and lower bounds true. [...] The results in Theorem 3 [...] hold for the more standard setting of bounded rewards.
>
> We note that, while for simplicity (and to be coherent with what it is commonly done in the literature) we proved Theorem 3 for bounded rewards in $[0,1]$, our analysis extends with almost no modifications to any sub-Gaussian distribution by simply changing the concentration thresholds (see footnote 3 in the revision for the precise definition). Therefore, our upper bounds when instantiated with such different thresholds (which would only change constant factors) **really match the lower bound of Theorem 1** for Gaussian rewards (up to a factor $H^2$ and logarithmic terms).
>
> > [...] in Theorem 1 there are no assumptions made about boundedness of the rewards [...] making the result similar in spirit to the lower bound in Theorem 4.5 (Dann et al. 2021)
>
> It is true that the lower bound (for regret minimization) in Theorem 4.5 of Dann et al. 2021 is the most related to ours. However, we point out that the PAC RL setting poses additional challenges which makes our proof depart significantly from the one of Dann et al. 2021. In particular, their proof is similar to only a small part of ours: the lower bound on the visitation counts for ($\epsilon$-)sub-optimal state-action pairs (Lemma 9). For those, one only needs to increase the reward of the considered pair by roughly its return gap to build an alternative MDP. This is enough to prove the entire lower bound for regret minimization. In our setting, we also need to consider all $\epsilon$-optimal pairs, for which the analysis is way more subtle, and deal separately with unique $\epsilon$-optimal pairs. Finally, we need to relate the total sample complexity to the local visitation lower bounds, yielding the novel formulation in terms of minimum flows.
>
> > Given that the paper does not really establish lower bounds in the more interesting setting of bounded rewards and the upper bounds mostly seem to remove horizon dependent factors, I am unsure of how strong the contributions of this work are.
>
> As mentioned above, our contributions go beyond merely improving horizon factors and we hope the reviewer will reconsider her/his position on this point. Moreover, as the understanding of instance optimality is quite limited in the MDP literature despite its importance, we believe that achieving it in the specific case of Gaussian rewards is alone a good contribution. How to do the same for bounded or general reward distributions remains an interesting open question for future work.

---

> > ### Author Response · Authors · 2022-08-01
> > **Answer to Reviewer vy1w (continued)**
> >
> > > On the fact that there could be a large gap between the complexity of bounded-reward MDPs and unbounded-reward ones
> >
> > We agree on this point. Indeed the two settings could have quite different complexities depending on the specific instance and on the reward bounds. Extending our lower bound for bounded rewards (and in general to any distribution class) is actually simple (we detail how to do that in response to the reviewer’s question below). However, we note that this would lead to quite inexplicit local lower bounds where the gaps would be replaced by the minimal KL divergence between the true reward distribution and one that yields an alternative MDP.
> > In this paper, our focus is mostly on deriving instance-dependent bounds scaling with the “standard” form of gaps defined as differences of expected values and characterizing when they are optimal. For the upper bounds, we can obtain such gaps for any sub-Gaussian distribution. However, for the lower bounds, it is known that such gaps are optimal only for Gaussian rewards. For instance, if we think about the (L)UCB algorithm for bandits, it is easy to get regret or sample complexity bounds which scale as a sum of inverse gaps for any sub-Gaussian distribution, though it is known that those are optimal only for Gaussian bandits. For general distributions, it is possible to derive lower bounds that scale with “KL gaps” as mentioned above, though different more involved algorithms are needed to match them (eg KL-UCB).
> >
> > > [...] will Algorithm 3 ever select a policy which visits the state-action pair with no incomming edges? Further, will Algorithm 1 play a policy which visits such a state-action pair?
> >
> > Both can indeed happen. In Appendix B.3, we report an example which shows that the minimum policy cover might need to play an eliminated arc. It is sufficient to imagine that the eliminated pair has a high number of “successors” that still need to be reached. Then, going through that pair might still be beneficial (or at least yield a cover of minimum size) for reaching them. Anyway, for the sample complexity of the algorithms, we only care about the number of policies in the cover (i.e., the number of episodes needed to visit everything) and not whether those policies go through eliminated pairs.
> >
> > > Do the authors think they can derive a meaningful lower bound, maybe in terms of a min-flow LP, for the setting of bounded rewards?
> >
> > Our lower bound can indeed be extended to general reward distributions (thus including bounded ones) and still yield a min-flow LP, though with an “implicit” KL-based definition of the gaps. We sketch how to do that. The main idea behind Lemma 7,8,9 is to build “alternative” MDPs by changing the mean reward at the single state-action pair under consideration. Now suppose we are given a general class of reward distributions $D$, i.e., $\nu_h(s,a) \in D$ for all $(s,a,h)$. If we want to build an alternative MDP which increases the reward at $(s,a,h)$ from $r_h(s,a)$ to $r_h(s,a)+\Delta$ (where $\Delta$ changes in the proofs of Lemma 7,8,9), we can just define the sub-set of distributions that achieve this property: $D_h(s,a) = (\nu’ \in D : E_{\nu’}[x] > r_h(s,a)+\Delta)$. Note that the set might be empty, in which case there is no alternative at $(s,a,h)$ as the reviewer suggested. Then, the local lower bound at $(s,a,h)$ is given by finding the closest (in KL divergence) distribution to $\nu$ from this set. That is, $E[n_h(s,a)] \geq \log(1/\delta) / \inf_{\nu’ \in D_h(s,a)} KL(\nu_h(s,a),\nu’)$, with the convention that the infimum over the empty set is plus infinity. We can repeat this argument for each $(s,a,h)$, essentially extending Lemma 7,8,9, to get a local lower bound for each triplet. Finally, the sample complexity lower bound is given as a minimum flow over these local lower bounds.
> >
> > Of course, whether such a lower bound is attainable by any algorithm (and how to do that) is an open question. It is likely that we need at least different concentration bounds based on KL divergences.

---

> > > ### Comment · Reviewer_vy1w · 2022-08-05
> > > **Response to answer**
> > >
> > > Thank you for the careful response. I agree that there is technical novelty in deriving both the upper and lower bounds and that there is added difficulty in extending instance dependent lower bounds from regret minimization to the PAC setting. However, I am unsure if I am miss understanding the provided argument regarding matching upper and lower bounds. My issue in particular is not with the fact that the lower bounds are stated for Gaussian rewards and I understand that it’s possible to extend the results in terms of the KL divergence.
> > >
> > > For simplicity assume that the rewards are Gaussian with variance 1. The issue I am trying to state is that if the expected rewards are bounded in [0,1] then there might not be a confusing MDP which is simply derived by perturbing a reward at a fixed (s,a)-pair by the desired gap, as the gap might be greater than 1. In this sense the class of MDPs with which you seem to be working in the upper bounds and the lower bounds section can be different as in the upper bounds section one works with the class of MDPSs with some fixed expected reward range while in the lower bounds this range can be H times larger to accommodate the appropriate confusing MDP. This can lead to example MDPs where the information theoretic upper bound for the restricted class (bounded expected rewards in [0,1]) is S times smaller than the lower bound for the larger class.

---

> > > > ### Author Response · Authors · 2022-08-05
> > > > **Response**
> > > >
> > > > We thank the reviewer for the follow up and for acknowledging the technical novelty of our contribution.
> > > >
> > > > **On matching upper and lower bounds**
> > > >
> > > > The reviewer is right that, if we derive the upper bound by assuming mean rewards to be bounded in [0,1], we cannot claim that our result matches the lower bound for Gaussian rewards (regardless of the analysis carrying out for unbounded distributions). However, we remark that we do *not* need such an assumption: our upper bound can be derived for any sub-Gaussian distribution *without* any condition on the mean rewards.
> > > >
> > > > The comment where we mention that we need mean rewards in [0,1] to carry out the analysis for sub-Gaussian distributions was present in the original version of the paper, but was removed in our revision (please see the new footnote 3). We will clarify this point in the final version and directly state Theorem 3 for the more general setting of sub-Gaussian distributions without any condition on their mean (as mentioned in footnote 3, the result is the same as the current one up to constants), while explicitly remarking that it matches (up to H^2 and logarithmic terms) the lower bound of Th. 1.
> > > >
> > > > We thank the reviewer for spotting this subtle and very important point.
> > > >
> > > > **On avoiding the assumption of mean rewards bounded in [0,1]**
> > > >
> > > > Finally, just a quick “technical” explanation of why the whole analysis carries out without assuming mean rewards to be bounded in [0,1]. Such an assumption is typically needed in stochastic MDPs to concentrate terms of the form
> > > >
> > > > $ \sum_{s’}(\hat{p}(s'|s,a) - p(s'|s,a)) V(s’)$
> > > >
> > > > where \hat{p}(s'|s,a) is our estimate of the transition probabilities and V(s’) is the value function of some policy (eg the optimal one). In order to use Hoeffding’s or Bernstein’s inequality to concentrate this term, and in order to use the resulting confidence intervals as bonuses in our algorithms, we need at least to know an upper bound on V, which is ensured by assuming a known upper bound on the (mean) rewards. In deterministic MDPs we do not need any of this: the return of each policy is simply a sum of H random variables (our rewards), and each can be concentrated by only assuming mild conditions on their distributions (eg sub-Gaussian), no matter what their mean is.

---

> > > > > ### Author Response · Authors · 2022-08-09
> > > > > **Follow up**
> > > > >
> > > > > We thank again the reviewer for engaging in the discussion with us.
> > > > >
> > > > > We wanted to follow up our last message to check whether there is any other issue that we can address. We believe that our responses already addressed the reviewer’s main concerns. In particular, we clarified that our result really matches the lower bound in the Gaussian case and that our contributions go beyond improving factors of the horizon from prior works.
> > > > >
> > > > > We hope the reviewer will reconsider their initial position and acknowledge our contributions with a higher score.

---

> > > > > > ### Comment · Reviewer_vy1w · 2022-08-10
> > > > > > **Comment**
> > > > > >
> > > > > > I am aware that the analysis for sub-Gaussian rewards does not require boundedness of the rewards as the concentration results use the sub-Gaussian norm. This is further made easier by the fact that concentration is needed only on the rewards and not the transition kernel. I am still not happy with the presentation, although I do agree that there is a setting in which the presented upper and lower bounds match and are tight. While I think this setting is okay to consider, the setting of expected rewards in [0,1] is more interesting as it would require non-trivial arguments which link the topology of the MDP to the possible confusing MDPs. This will happen because one cannot just perturb the reward of a fixed state-action pair in the original MDP in order to obtain a confusing MDP as the gap in value functions might exceed 1.
> > > > > >
> > > > > > I am willing to raise my score, however, the reviewers should make sure to include a careful discussion regarding the case when the upper and lower bounds match, i.e., mean rewards need not be bounded in [0,1].

---

> > > > > > > ### Author Response · Authors · 2022-08-10
> > > > > > > **Follow up**
> > > > > > >
> > > > > > > We thank the reviewer for the quick answer.
> > > > > > >
> > > > > > > In the final version, we will certainly include a precise discussion on the settings where lower and upper bound match (i.e., Gaussian rewards with no restriction on their mean) and also give some hints on how to refine our results if we assume mean rewards to be bounded in [0,1].

---

### Official Review · Reviewer_ELtb · 2022-07-11

**Rating:** 6
**Confidence:** 3
**Soundness:** 3 good
**Presentation:** 3 good
**Contribution:** 3 good

**Summary:**

This paper studies PAC RL in tabular deterministic MDPs with known transitions but unknown (stochastic) rewards. An instance-dependent lower bound is presented which bounds the sample complexity in terms of the solution to a min-flow problem where the edge function is related to (the inverse) of a newly-defined deterministic return gap. A new algorithm is presented (EPRL) which is worst-case minimax optimal and matches the instance-dependent lower bound up to an H^2 term and logarithmic factors.

**Questions:**

- How much is the assumption that transitions are known matter? It seems like it is mainly used in Line 7 of the algorithm (to calculate Pi_{s,a,h}). Can the algorithm be modified or this assumption be removed?

**Limitations:**

Limitations: See "Strengths and Weaknesses" section above.

Potential negative societal impacts: N/A

**Strengths And Weaknesses:**

Significance: There has been quite a lot of work lately on instance-dependent results for RL. This paper adds nicely to that body of work. Unfortunately, the setting is restricted to known deterministic transitions, and there is quite a large gap between the lower bound and the upper bound (H^2, ignoring logarithmic factors). As a first instance-dependent PAC result, this is still a nice result. The experimental results and the discussion is interesting as well. For want of time I have not checked the proofs for any mistakes.

Originality: The connections with minimum flow are interesting. The EPRL algorithm is novel.

Clarity: The paper is well written overall. One complaint is that, since the upper bounds and the lower bounds differ by a factor of H^2, the claims that the algorithm nearly matches the lower bound is misleading and should be modified. Another misleading claim is that the transition function can be learned "with only an additional constant" (this constant would give an additional dependence on SAH). Lastly, the claim that this is "the first instance-dependent lower bound for the PAC setting in episodic MDPs" is technically true, but there are some instance-dependent bounds for best-policy identification in the cited papers [44] and [4] (a restricted case of PAC), as well as several instance-dependent regret bounds in cited papers [13] and [40] which should be mentioned.

Smaller complaints:
- The notation and naming for the different instance-dependent functions are quite similar to each other, making it difficult to easily remember which is which (e.g. value gap, return gap, deterministic return gap, and the notations  \Delta, \tilde\Delta, \bar\Delta).
- The definition of Pi^t should be included in the pseudo-code of Algorithm 1 for readability

Typos:
- Line 182 "mimixal optimal"

---

> ### Author Response · Authors · 2022-08-01
> **Answer to Reviewer ELtb**
>
> We thank the reviewer for the positive feedback and for the suggestions on improving the presentation. We address all the questions below.
>
> > [...] since the upper bounds and the lower bounds differ by a factor of $H^2$, the claims that the algorithm nearly matches the lower bound is misleading and should be modified.
>
> We thank the reviewer for pointing this out. In our revised draft, we made sure that every time we say “nearly matching” we specify “up to a factor of $H^2$ and logarithmic terms”. Our main focus has been on identifying a meaningful notion of gaps (i.e., a key problem-dependent quantity) appearing both in the lower and the upper bounds, neglecting the optimal dependency in the horizon (a problem-independent quantity). About the latter, see the new remark after Theorem 3 and the answer to Reviewer SFoB for a discussion on how to achieve optimality. We remark that other instance-dependent upper bounds obtained for either regret or sample complexity in episodic MDPs also feature some possibly suboptimal multiplicative factors in (powers of) H, see e.g. [36], [44], [13].
>
> > Another misleading claim is that the transition function can be learned "with only an additional constant" (this constant would give an additional dependence on SAH).
>
> The reviewer is right that this additional sample complexity is not constant in all problem variables (our claim only refers to \epsilon and \delta), so we have rephrased that sentence accordingly.
>
> > How much is the assumption that transitions are known matter? [...] Can the algorithm be modified or this assumption be removed?
>
> The assumption of known transitions is used to compute upper and lower bounds on the Q-values, which are useful for checking eliminations (Line 7 of Algorithm 1). It is also used to calculate the set $S_h$ of reachable states in step $h$, in which we initialize the set of candidate actions, and indeed to calculate $\Pi_{s,a,h}$. If the transitions are unknown, we can trivially modify the algorithm by adding an initialization phase to discover the dynamics of the MDP which takes at most SAH episodes.
>
> > [...] there are some instance-dependent bounds for best-policy identification in the cited papers [44] and [4] (a restricted case of PAC), as well as several instance-dependent regret bounds in cited papers [13] and [40] which should be mentioned.
>
> We added a pointer to the regret lower bounds presented in [13] and [40]. To the best of our understanding, [4] is not considering the episodic setting and [44] is not providing an instance-dependent lower bound. They provide a worst-case result saying that there exists an MDP for which the complexity of any algorithm minimizing regret is larger than that of the MOCA algorithm. This does not provide a lower bound on the sample complexity of any ($\epsilon,\delta$)-PAC algorithm.
>
> > The definition of $Pi^t$ should be included in the pseudo-code of Algorithm 1 for readability
>
> We have updated the pseudo-code by adding the definition of $\Pi^t$ as the reviewer suggests.

---

> > ### Comment · Reviewer_ELtb · 2022-08-09
> > **Thanks for your reply**
> >
> > Thanks to the authors for their reply!
> >
> > I felt largely positive about this paper and this has only improved after the writing improvements which the authors are making.

---

### Official Review · Reviewer_n7wX · 2022-07-11

**Rating:** 7
**Confidence:** 5
**Soundness:** 4 excellent
**Presentation:** 3 good
**Contribution:** 4 excellent

**Summary:**

This paper studies instance-dependent optimality of PAC reinforcement learning in tabular deterministic episodic MDP.
It first introduces a new notion of sub-optimality gap called deterministic return gap, then characterizes an instance-dependent lower bound in terms of this new definition. The paper also proposes an algorithm based on maximum coverage sampling and confidence-based elimination, and prove that it can match the lower bound up to a $H^2$ factor. Finally, the paper also provides empirical evaluation on synthetic MDPs.



**Questions:**


Here are some question that I hope the authors can answer in the rebuttal.

1. There is a factor $H^2$ gap between upper and lower bound. It seems to be the mismatch is caused by the lower bound as it is hard to get explicit dependence on the $H$ factor (because the lower bound is between max and sum over all horizons). Is there a room to improve this?

2. How can the lower bound be generalized to the stochastic setting? I guess most part of the analysis (or the high level strategy) will remain unchanged. Basically one only needs to argue for the KL decomposition used in the proofs in Section C. But I am not sure if there are other difficulties.

I also have some small suggestions.

1. The notations can be improved somehow. For example, what's $n^\tau_h(s,a)$ in equation (3)? What is $\hat{pi}$ in line 727? Maybe all of these definitions are given somewhere in the paper, but it's hard for the readers to check back and forth to find them. After reading carefully I can understand the meaning of the notations, but it will be very helpful if the authors can add a some explanation in the proof.

2. Lemma 1 from Kaufmann et al., 2016 has been used quite often in the proof of lower bound. Maybe it's helpful to include that in the appendix for completeness?

3. Some discussion in the main paper about how to prove instance-dependent lower bound will be useful.

**Ethics Review Area:**

["I don’t know"]

**Limitations:**

Please see the above section.

**Strengths And Weaknesses:**

Strength:

I vote for acceptance as this paper presents solid contributions and novel proving techniques.

Instance-dependent analysis is a very important topic in RL theory and this paper provides some really contributions. In general I really enjoying reading this paper. One of the main contribution is to include graph-theoretical concepts in the lower bound analysis: it first proves a lower bound on minimum number of visitation for each state-action pair, the instance-dependent lower bound follows immediately by realizing that the visitation counts follows a feasible flow. I think this novel technique also has potential application in  other analysis of RL problems.

Weakness:

Some notations and definitions can be improved.

---

> ### Author Response · Authors · 2022-08-01
> **Answer Reviewer n7wX**
>
> We thank the reviewer for the positive feedback and for the suggestions on improving the presentation. We address all the questions below.
>
> > There is a factor $H^2$ gap between upper and lower bound. [...] Is there a room to improve this?
>
> We believe that the lower bound could be tightened to feature a $H$ multiplicative factor, but so far we were only able to achieve so for the specific case of tree-based MDPs, see the lower bound proved in Appendix E and Remark 2 in our revised paper. We also refer the reviewer to the answer to Reviewer SFoB for further details on the optimal dependence on $H$.
>
> > How can the lower bound be generalized to the stochastic setting?
>
> After the submission, we managed to prove a lower bound in the stochastic setting by indeed using the same construction (changing a single reward in the MDP so as to change the set of optimal policies), which features a different, intriguing notion of gap. We are still investigating whether this lower bound can be matched, so the stochastic case still requires some work.
>
> > What is $n_h^\tau(s,a)$ in equation (3)? What is $\widehat{\pi}$ in line 727?
>
> $n_h^\tau(s,a)$ denotes the number of visits to (s,a,h) at the stopping time \tau. We use \widehat{pi} to denote the policy returned by the recommendation rule at stopping (see the paragraph “learning problem” in Section 2 for the definition). We added an explanation at line 727 as the reviewer suggests.
>
> > Maybe it's helpful to include [Lemma 1 of Kaufmann et al., 2016] in the appendix for completeness?
>
> Thanks for this suggestion. In our revision, we added a paragraph at the beginning of Appendix C presenting the change of distribution lemma which is repeatedly used.
>
> > Some discussion in the main paper about how to prove instance-dependent lower bound will be useful.
>
> We will add a brief proof sketch in the final version, as the 9 page limit prevents us from doing so at the moment.

---

### Official Review · Reviewer_SFoB · 2022-07-12

**Rating:** 7
**Confidence:** 3
**Soundness:** 3 good
**Presentation:** 3 good
**Contribution:** 3 good

**Summary:**

This paper studies PAC reinforcement learning, where an agent is required to identify an $\epsilon$-optimal policy with probability $1-\delta$. The instance-dependent complexity remains elusive in episodic Markov decision processes (MDPs). In this paper, the nearly matching upper and lower bounds on the sample complexity of PAC RL in deterministic episodic MDPs with finite state and action spaces. In particular, our bounds feature a new notion of sub-optimality gap for state-action pairs that we call the deterministic return gap. The design and analyses employ novel ideas, including graph-theoretical concepts (minimum flows) and a new maximum-coverage exploration strategy.

**Questions:**

I have already provided all the comments in above.

**Limitations:**

Yes.

**Strengths And Weaknesses:**

**Strgenths** This paper provides fine-grained solution for PAC reinforcement learning with the deterministic MDP setting. The Proposed algorithm is based on the elimination rule and the resulting complexity is expressed in terms of min flow with linear program formulation. The experiment is motivate.

**Weakness**

- The claimed optimality actually differs from the upper bound by a factor of $H^2$, therefore the gap is still there. While notice the discussion about this from 327-332, I still wondering if it is possible to improve horizon dependence for EPRL? Current analysis seems based on Hoeffding's technique eon(5), could some (empirical) Bernstein bonus be designed  for EPRL?

- The key procedure for EPRL is the policy elimination. How do you conduct efficient procedure for this elimination step to make algorithm practical? Especially the policy set can be exponential large?

- Deterministic MDP is a simpler setting comparing to stochastic MDPs since each state action only needs to be experienced and the problem size is $SA$ (instead of $S^2A$ for the stochastic case). Hence, it is expected to have faster convergence than the general tabular MDP problem. How do you interpret the $O(\frac{SAH^2}{\epsilon^2})$ sample complexity as the optimal complexity for deterministic MDP? Is there a minimax lower bound for that? Indeed, in the offline case, it has been noted learning deterministic MDPs can have faster convergence with rate $\frac{1}{n}$ instead of the standard statistical rate $\sqrt{\frac{1}{n}}$ due to its instance dependent nature [Yin&Wang,21]. I believe it is very necessary to refer [Yin&Wang,21] and discuss potential connections to the current work, for example, can instance-dependent online RL also achieve smaller regret comparing to the standard $\sqrt{T}$?

[Yin&Wang,21] Towards Instance-optimal Offline Reinforcement Learning with Pessimism, NeurIPS 21.

---

> ### Author Response · Authors · 2022-08-01
> **Answer to Reviewer SFoB**
>
> We thank the reviewer for the positive feedback. We address all the questions below.
>
> > The claimed optimality actually differs from the upper bound by a factor of $H^2$ [...]
>
> Our main focus was on identifying a meaningful notion of gaps appearing both in the upper and lower bound, leaving open the optimal dependency in the horizon. We actually conjecture that this gap could be reduced to a single $H$ factor by boosting the lower bound. While we have not been able to prove it for the general case, we provided a lower bound with an improved $H$ factor for the special case of tree-based MDPs in Appendix E. To improve the upper bound, as the reviewer suggests, it is indeed likely that we need tighter concentration inequalities over value functions. We added a remark about this after Theorem 3 in the revised draft.
>
> > [...] could some (empirical) Bernstein bonus be designed for EPRL?
>
> To elaborate a little on what we need, $H^2$ actually comes from upper bounding the diameter $\sum_{h=1}^{H}1/\sqrt{n_h^t(s_h^\pi,a_h^\pi)}$ by $H$ times the largest bonus. With some tighter concentration bounds, we should improve the diameters to $\sqrt{\sum_{h=1}^{H}1/n_h^t(s_h^\pi,a_h^\pi)}$, which would indeed shave off a factor $H$. However, we believe that *Bernstein inequality alone would not be enough* to achieve this (after all, in the Gaussian case, all rewards have a fixed positive variance). It is likely that we need a bound on the cumulative deviation of the reward estimator across multiple $(s,a,h)$ triplets rather than one independent bound for each $(s,a,h)$. We will explore this direction in future work.
>
> > The key procedure for EPRL is the policy elimination. How [...] to make algorithm practical?
>
> While EPRL’s pseudocode features a set of active policies $\Pi^{t}$, we clarify in our revision (see Remark 1) that the algorithm is never actually storing/enumerating this set of policies. In particular, **EPRL does not eliminate policies** but rather state-action-stage triplets which are not likely to be crossed by any optimal policy (which leads to an update of the set of candidate actions $\mathcal{A}_h^{t}(s)$ for each state, which leads to an implicit update of $\Pi^{t}$).
>
> Checking for eliminations requires computing at most $SAH +1$ dynamic programs (DPs): 1 DP to compute $\max_\pi \underline{V}_1^{\pi}(s_1)$ and 1 DP for each $(s,a,h)$ to compute
>
> $\max_{\pi \in \Pi_{s,a,h} \cap \Pi^{t-1}} \overline{V}_1^{\pi}(s_1)$.
>
> Checking for stopping requires to compute the maximum over $\Pi^{t}$ of $\sum_{h=1}^{H}b_h^{t}(s_h^{\pi},a_h^{\pi})$ which can also be computed efficiently using dynamic programming. Overall, the computational complexity of our algorithm is polynomial in $SAH$ (see the paragraph “computational aspects” in Section 7). Finally, we note that to reduce the computational cost, one can perform eliminations only rarely (e.g. every $10 SAH$ episodes as we did in our experiments or even at exponentially-separated phases, like when the number of episodes doubles) without compromising the theoretical results.
>
> > How do you interpret the [...] optimal complexity for deterministic MDP? Is there a minimax lower bound for that?
>
> The $O(SAH^2/\epsilon^2)$ sample complexity of EPRL is indeed *minimax optimal* for deterministic MDPs, since we proved a matching worst-case lower bound in Appendix C.2 (Theorem 7, mentioned in the last paragraph of Section 3 in the main paper).
>
> > [...] it has been noted learning deterministic MDPs can have faster convergence [...] [Yin&Wang,21].
>
> As noted by the reviewer, the optimal minimax sample complexity exhibits a reduced $H$ factor compared to the general fully stochastic case. We think that in the regret setting, we could similarly prove $\sqrt{H^2SAT}$ matching upper and lower bounds which are smaller than the general $\sqrt{H^3SAT}$ obtained for fully stochastic MDPs. We thank the reviewer for pointing out the paper of Yin and Wang, 2021. We added a reference to it in the revision as another example in which we can get faster rates for MDPs with deterministic transitions (see the last paragraph of Section 3).

---

### Meta-Review · Area_Chair_kT3K · 2022-08-25

**Recommendation:** Accept
**Confidence:** Certain

**Metareview:**

The paper studies PAC reinforcement learning in tabular episodic MDPs with deterministic transitions and provides upper and lower bounds on the sample-complexity that match up to horizon and log-factors.
Overall, all reviewers rate this paper positively (after the authors' responses and discussion). They view the contribution of a fine-grained instance-dependent guarantees in this setting as significant and particularly appreciated the novel insights, e.g., relating the MaxCoverage function in Algorithm 1, to the StaticMaxCoverage in Algorithm 3 or the inclusion of graph-theoretical concepts in the lower bound analysis. There were also several limitations raised, in particular the deterministic transition assumption and the reward range assumption used in the lower bound. However, some of these can be addressed by clarification and more detailed discussion in the camera ready. All in all, this is a solid paper and is recommended to be accepted.

**Award:**

No

---

### Decision · Program_Chairs · 2022-09-14

Accept